# Use of fourth-generation rapid combined antigen and antibody diagnostic tests for the detection of acute HIV infection in a community centre for men who have sex with men, between 2016 and 2019

**Jorge Saz**[1]*, **Albert Dalmau-Bueno**[2], **Michael Meulbroek**[1], **Ferran Pujol**[1], **Josep Coll**[3], **Ángel Herraiz-Tomey**[4], **Félix Pérez**[1], **Giovanni Marazzi**[1], **Héctor Taboada**[1], **Dante R. Culqui**[4,5], **Joan A. Caylà**[6], **BCN Checkpoint Working Group**¶

1 Projecte dels NOMS-Hispanosida, Barcelona, Spain, 2 BCN Checkpoint, Barcelona, Spain, 3 IrsiCaixa AIDS Research Institute, Fight AIDS and Infectious Diseases Foundation, Badalona, Spain, 4 Grupo Pulso (a company of Evidenze Group), Sant Cugat del Vallés, Spain, 5 Isabel Roig-Blauclinic Socio-Sanitary Centre, Barcelona, Spain, 6 Tuberculosis Research Unit of Barcelona (UITB) Foundation, Barcelona, Spain

¶ The complete membership can be found in Acknowledgments.
* jsaz@hispanosida.com

## Abstract

### Objective

To assess the use of fourth-generation rapid diagnostic tests in identifying acute infection of Human Immunodeficiency Virus (HIV).

### Methods

BCN Checkpoint promotes sexual health among men who have sex with men (MSM), with a focus on diagnosing HIV early, initiating combined antiretroviral treatment (cART) promptly, and recommending regular repeat testing for those who have tested negative. This cross-sectional study included all test results obtained at the centre between 25 March 2016 and 24 March 2019. The *Alere™ HIV Combo* (now rebranded to *Determine™ HIV Ultra, from Abbott*) was used to detect p24 antigen (p24 Ag) and/or immunoglobulin M (IgM) and G (IgG) antibodies to HIV-1/HIV-2 (HIV Ab). Rapid polymerase chain reaction (PCR) confirmatory testing and Western blot (WB) were performed for clients with a positive rapid test result. Confirmed HIV cases were promptly referred to the HIV unit for care and cART prescription.

### Results

A total of 12,961 clients attended BCN Checkpoint during the study and 27,298 rapid tests were performed. 450 tests were found to be reactive, of which 430 confirmed as HIV-positive, representing a prevalence of 3.32%. Four confirmed cases (0.93%) were detected as "p24 Ag only", nine (2.09%) as "both p24 and HIV Ab" and 417 (96.98%) as "HIV Ab only".

**Data Availability Statement:** All relevant data are within the paper and its Supporting Information file.

**Funding:** Abbott Laboratories has financed the editing and publishing of the results. However, the company had no role in study design, data collection and analysis, decision to publish, or preparation of the manuscript.

**Competing interests:** We would like to declare explicitly that none of the authors or members of the BCN Checkpoint Working Group have or had during the last five years any financial or nonfinancial competing interests with the company Abbott Laboratories. Therefore, this does not alter our adherence to PLOS ONE policies on sharing data and materials.

The "p24 Ag only" group had a 1-log higher viral load than the other groups and initiated treatment on the following working day. Overall, there were 20 false-positive results (0.07% and 4.44% of total and reactive tests, respectively), of which 10 positive for "p24 Ag only" and 10 for "HIV Ab only".

## Conclusions

Four Acute HIV Infections (AHI), with very high viral loads, have been detected with the "p24 Ag only" while the HIV Ab were still absent. Referral to the HIV unit and initiation of cART on the following working day contributed to improving persons' health and to reduce HIV transmission chain.

## Introduction

The HIV epidemic has become an endemic infection. It is one of the world's most serious public health challenges that need a global commitment to stopping new HIV infections and ensuring that everyone has access to HIV treatment. According to the UNAIDS (Joint United Nations program on AIDS/HIV), in 2018, 37.9 (32.7–44.0) million people were living with HIV worldwide: 36.2 (31.3–42.0) million were adults and 1.7 (1.3–2.2) million were children under the age of 15 [1].

In the last three decades, over 2.3 million people have been diagnosed with HIV in the European region, according to World Health Organization estimates [2]. In 2017, a total of 159,429 individuals were newly diagnosed in 50 out of 53 countries of the European region, which equals a rate of 20 new HIV diagnoses per 100,000 residents.

HIV is known to affect mainly key populations in urban areas. In San Francisco, one of the main gay-friendly cities, in 2017 a total of 15,990 people were estimated to be living with HIV, or 1.81% of the population. During that same year, another 197 people were newly diagnosed with HIV, of which 63% were men who have sex with men (MSM) [2]. In that same year in London 33,436 people were estimated to be living with HIV and the number of new HIV diagnoses was 1,549, of which 63% were MSM [3]. Finally, in 2017 in Barcelona, 352 cases were detected, of which 82.5% were MSM [4].

With the introduction of new antiretroviral therapies, the WHO promoted the test-and-treat strategy in 2009 [5]. Later on, UNAIDS launched the 90–90–90 targets as part of its proposed goals for 2020. The aim was to have 90% of all people living with HIV to know their HIV status, 90% of all HIV-diagnosed persons to receive sustained antiretroviral therapy, and 90% of all people on HIV treatment to achieve viral suppression [6]. However, the biggest gap can be found in the first 90% target (diagnosis). On the other hand, some studies show that, due to the very high viral load during the initial infection stage, 50–70% of new infections are produced as a consequence of acute HIV infections [7]. For this reason, it is important not only to detect new infections, but also to detect as early as possible, preferable within the first days of the infection.

The use of fourth-generation rapid HIV tests allows for simultaneous detection of p24 Ag and HIV Ab. This is an important step towards early diagnosis of HIV infection [8]. The p24 Ag is a marker for early infection and can be detected between 15 and 22 days post-exposure to HIV and before antibodies are detectable [9]. The p24 antigen detection is transient because, as antibodies begin to develop, they bind to the p24 antigen and form immune complexes that interfere with p24 assay detection, unless the assay includes steps to disrupt the antigen-

antibody complexes. Next, immunoglobulin M antibodies are expressed, which can be detected by 3rd and 4th generation immunoassays 3 to 5 days after p24 antigen is first detectable, and 10 to 13 days after the appearance of viral RNA. Therefore fourth-generation tests can detect both early and non-early infections [10, 11].

After the infection has taken place, over 50% of individuals will develop an acute retroviral syndrome with nonspecific features [12]. In practice they usually go undiagnosed and are easily mistaken as other viral infections like infectious mononucleosis or influenza. If HIV is detected during the acute phase, cART suppresses the symptoms of acute viremia, reduces viral reservoirs and improves the long-term outcome of individuals, as well as a significant reduction of viral transmission [12, 13].

HIV infection during the acute stage is more transmittable than during later stages of the infection, due to the high levels of viral load [13]. Therefore, precise, timely detection of primary HIV infection is critical, as it is an opportunity to start treatment early and improving the future health of the individual, as well as to interrupt the transmission chain and reaching the first 90% target.

Providing access to diagnosis is important and implies prioritising of 1) facilitating low barrier and timely testing [14, 15] and 2) providing prompt access to cART, with the aim of reducing the community viral load [16–21].

The aim of this article is to assess the added value of fourth-generation testing in diagnosing acute HIV infection in a community centre for MSM and transgender women (TW).

## Methodology

### Type of study

Analytical cross-sectional. The study protocol was approved by the Clinical Research Ethics Committee of the Hospital Germans Trias i Pujol and all the participants provided their informed consent for participating.

### Study population

All clients who attended BCN Checkpoint in the city of Barcelona during the period between 25 March 2016 and 24 March 2019 were enrolled in the study. BCN Checkpoint is a community centre for the detection of HIV and other sexually transmitted infections, targeted at gay men, other MSM and TW. BCN Checkpoint was founded in 2006 in the gay area of Barcelona and is characterised by a peer-led focus and the use of Point-of-Care (PoC) technology. Since 2014, BCN Checkpoint has started campaigns to raise awareness about regular testing, as regular as every three months in case of having sex with several partners and anal intercourse without a condom.

### Classification of infections

According to the definition offered in the Department of Health and Human Services (DHHS) guidelines for the use of antiretroviral therapy in the treatment of HIV infection [22], there are three different infection statuses: 1) **acute infection** is detected by the presence of HIV RNA (ribonucleic acid) or p24 Ag, but not HIV ab; 2) **recent infection**, in which case both RNA and HIV Ab tests are positive, suggesting the infection occurred in the last six months, and 3) **early infection**, representing either acute or recent infection.

Regarding late and advanced presentations, the definitions agreed upon by the European Late Presenter Consensus Working Group were used. **Late presentation with HIV disease/ infection:** people presenting with a CD4 count below 350 cells/μL or an AIDS-defining event,

regardless of the CD4 cell count. **Presentation with advanced HIV disease/infection:** people presenting for care with a CD4 count below 200 cells/μL or an AIDS-defining event/disease, regardless of the CD4 cell count [23].

## Types of testing employed, characteristics and interpretation

Alere™ HIV Combo (rebranded to Determine™ HIV Ultra, from Abbott) was used for the screening of HIV infection. It is a fourth-generation rapid test that detects IgM and IgG Abs to HIV-1/HIV-2. Moreover, it detects p24 Ag to HIV-1. This is what sets it apart from third-generation testing, as it allows detecting both HIV Abs and p24 Ag with the same test [24].

Rapid, qualitative HIV RNA testing (Xpert® HIV-1 Qual, Cepheid) was performed to confirm reactive p24 Ag and/or HIV Ab test. If result detected HIV, a positive HIV diagnosis was confirmed. In case of divergent results, a second quantitative HIV RNA test (Xpert® HIV-1 Viral Load, Cepheid) was performed. If the second quantitative HIV RNA test result was positive, the final HIV diagnostic result was considered positive, whereas if the second test was negative, the final HIV diagnosis was negative. In parallel a blood sample was send to an external laboratory to perform a Western Blot assay (WB). Tests that produced false-positive results for the "p24 Ag only" were investigated by the manufacturer.

A rapid testing of T-lymphocyte counts (PIMA™ CD4, Abbott) was also performed to assess baseline immunity.

## Statistical analysis

**Study variables.** A descriptive analysis of the main demographic features among MSM and TW was performed, including age, level of education, country of birth, and number of clients and tests. Variables recorded included the dates for testing or follow-up testing, and the date of cART initiation in HIV-infected individuals with acute HIV infection. The time between visits was calculated based on these variables. This time period was also disaggregated by number of sexual partners. The time elapsed between visits, derived from the registered dates (detection visit and cART initiation visit), was analysed. Other variables studied included the results of p24 Ag and HIV Ab testing, viral load, and CD4 counts.

The description of these variables was stratified according to the confirmatory result (confirmed or reactive/no-confirmed or nonreactive). A chi-squared test (or Fisher's exact test) and a Student's t-test (or Wilcoxon signed-rank test) were employed to analyse any possible associations between qualitative and quantitative variables, respectively. A p-value under 0.05 was regarded as statistically significant. A 95% confidence interval (95% CI) was also calculated for all of the percentages.

For the analysis, Stata 14 software was used.

**Ethical considerations.** All participants gave their verbal and written informed consent to get tested. All users received peer counselling, and those who tested positive for HIV received emotional support by peers and adequate information, they were offered direct and fast referral to the HIV unit for care and to initiate cART promptly. Any information that could personally identify the participant was excluded.

## Results

During the three-year study period 12,961 clients, without a previous HIV diagnosis, attended the centre and a total of 27,298 rapid tests were performed. The average age was 33.37 years (95% CI: 33.20–33.54); 67.76% belonged to the 25–44 age group; 62.46% had higher (university) education; and 55.61% were born in Spain. The average time elapsed between a negative test and the follow-up test was 7.03 months (95% CI: 5.86–6.04) for all clients. According to

the number of sexual partners during last six months differences were observed for those who had less than two sexual partners (8.65 months, 95% CI: 8.33–8.97); between two and nine partners (7.29; 95% CI: 7.13–7.45); and those with 10 or more partners (6.09 months; 95% CI: 5.93–6.25) (Table 1).

From the 27,298 rapid tests performed to 12,961 different clients, a total of 450 reactive test results (1.65% of tests and 3.47% of persons) were obtained, of which 430 confirmed to be positive (1.58% and 3.32% respectively). Of the reactive tests 20 (0.07% and 0.15% respectively) resulted to be a false-positive result (Table 2, Fig 1). The highest prevalence of confirmed cases was observed in the 25–34 age-group, with an unknown or primary level of education, and Latin America for country of birth (Table 1). In addition, the 25–34 age group accounts for half of the confirmed infections.

**Table 1. Prevalence of HIV infection according to demographic features and time elapsed between visits in participants that got tested at BCN checkpoint, Spain (25 March 2016 to 24 March 2019).**

| Study variables | Confirmed positives | Confirmed negatives | Tested total | p-value | Prevalence (95% CI) |
|---|---|---|---|---|---|
| Number of HIV performed tests | | | 27 298 | NA | |
| Number of persons tested | 430 (3.32%) | 12 531 (96.68%) | 12 961 | NA | 3.55 (3.24–3.89) |
| Average age[a] | 33.22 (32.41–34.03) | 33.37 (33.20–33.54) | 33.37 (33.20–33.54) | 0.155 | |
| **Age (years)** | | | | **0.008** | |
| Under 25 years | 53 (12.33%) | 2256 (18.0%) | 2309 (17.81%) | **0.0025** | 2.30 (1.72–2.99) |
| 25–34 years | 215 (50.0%) | 5304 (42.33%) | 5519 (42.58%) | **0.0016** | 3.90 (3.4–4.44) |
| 35–44 years | 114 (26.51%) | 3150 (25.14%) | 3264 (25.18%) | 0.5187 | 3.49 (2.89–4.18) |
| 45 years and above | 47 (10.93%) | 1694 (13.54%) | 1741 (13.43%) | 0.1217 | 2.70 (1.99–3.57) |
| Unknown | 1 (0.23%) | 127 (1.01%) | 128 (0.99%) | 0.1731 | 0.78 (0.02–4.28) |
| **Level of education** | | | | **0.119** | |
| Unfinished primary education | 1 (0.23%) | 13 (0.10%) | 14 (0.11%) | 0.9665 | 7.14 (0.18–33.87) |
| Primary education | 14 (3.26%) | 291 (2.32%) | 305 (2.35%) | 0.233 | 4.59 (2.53–7.58) |
| Secondary education | 147 (34.19%) | 3812 (30.42%) | 3959 (30.55%) | 0.1244 | 3.71 (3.15–4.35) |
| Higher education | 241 (56.05%) | 7678 (61.27%) | 7919 (61.10%) | **0.0096** | 3.04 (2.68–3.45) |
| Unknown level of education | 27 (6.28%) | 737 (5.88%) | 764 (5.89%) | 0.088 | 3.45 (2.25–5.06) |
| **Country of birth** | | | | **<0.001** | |
| Spain | 182 (42.33%) | 7016 (56.03%) | 7208 (55.61%) | **0.000** | 2.52 (2.18–2.91) |
| Western Europe | 51 (11.86%) | 1846 (14.74%) | 1901 (14.67%) | **0.0107** | 2.68 (2.00–3.51) |
| Central and South America (Latin America) | 143 (33.26%) | 2203 (17.59%) | 2350 (18.13%) | **0.000** | 6.09 (5.15–7.13) |
| Other origins | 44 (10.23%) | 1122 (8.96%) | 1157 (8.93%) | **0.000** | 3.80 (2.78–5.07) |
| Unknown country of birth | 10 (2.33%) | 334 (2.67%) | 345 (2.66%) | 0.1887 | 2.90 (1.40–5.27) |
| **Time (months) elapsed between visits, disaggregated by the number of sexual partners in the last six months, average (95% CI)** | | | | | |
| Total | 7.05 (5.87–6.05) | 6.08 (5.12–6.52) | 7.03 (5.86–6.04) | **0.030** | NA |
| Less than two partners | 8.65 (8.32–8.96) | 8.41 (5.17–12.41) | 8.65 (8.33–8.97) | 0.886 | NA |
| Between 2–9 partners | 7.29 (7.13–7.45) | 7.81 (5.42–8.8) | 7.29 (7.13–7.45) | 0.555 | NA |
| 10 partners or more | 6.11 (5.94–6.26) | 5.11 (4.75–6.33) | 6.09 (5.93–6.25) | 0.054 | NA |
| Don't know/Don't answer | 7.88 (6.88–8.72) | 4.78 (4.02–9.56) | 7.78 (6.88–8.68) | 0.240 | NA |

a 95% CI. b Time in months, day average (95% CI). HIV: human immunodeficiency virus; 95% CI: 95% confidence interval; NA: nonapplicable.

**Table 2. Results of antigen and/or antibody rapid tests performed at BCN checkpoint, Spain (25 March 2016 to 24 March 2019).**

|  | p24 Ag only | both p24 + HIV Ab | HIV Ab only | Total |
|---|---|---|---|---|
| **Rapid HIV test results** | **n (%)** | **n (%)** | **n (%)** | **n (%)** |
| Reactive tests total | 14 (3.11) | 9 (2.00) | 427 (94.89) | 450 (100) |
| False positives | 10 (50.00) | 0 (0.00) | 10 (50.00) | 20 (4.44) |
| **Confirmed positives** | **4 (0.93)** | **9 (2.09)** | **417 (96.98)** | **430 (95.56)** |

Of the 430 HIV-positive confirmed cases, four cases (0.93%) were diagnosed by the presence of the "p24 Ag only", nine (2.09%) by "both p24 Ag and HIV Ab" and 417 (96.98%) by "HIV Ab only". A total of 13 tests included the p24 Ag in the result. Regarding the 20 false-positive results, 10 cases were observed with the positive "p24 Ag only" (10/14, 71.43%) and another 10 (10/427, 2.34%) with a positive "HIV Ab only" result (Table 2, Fig 1). Plasma and serum samples of seven out of the ten false-positive "p24 Ag only" results were sent to the manufacturer for further investigation and were found to fall in the established ranges of test performance for each affected batch of the kit.

The positive predictive value (PPV) of the tests can be calculated from the data in Table 2. In case of "p24 Ag only" the PPV was 28.57%; for "both p24 Ag and HIV Ab" the PPV was 100.00%; and for "HIV Ab only" 97.66%.

When comparing the initial viral load of the 4 confirmed cases of "p24 Ag only" to the 9 confirmed cases of "both p24 Ag and HIV Ab" results showed that the first group had a 1-log higher viral load than the second (Table 3). Therefore the "p24 Ag-only" group might have a higher probability of an HIV transmission and were offered a fast referral, within 24 hours, with an immediate start of cART. The time elapsed between diagnosis and cART initiation for each group was (Table 3). The "p24 Ag only" group initiated cART on the following working day. Of the "both p24 Ag and HIV Ab" group, one case arranged his own referral; average time for referral was 6.75 days; 4/8 (50.00%) started cART the same day of the hospital visit and 3/8 (37.50%) within a week.

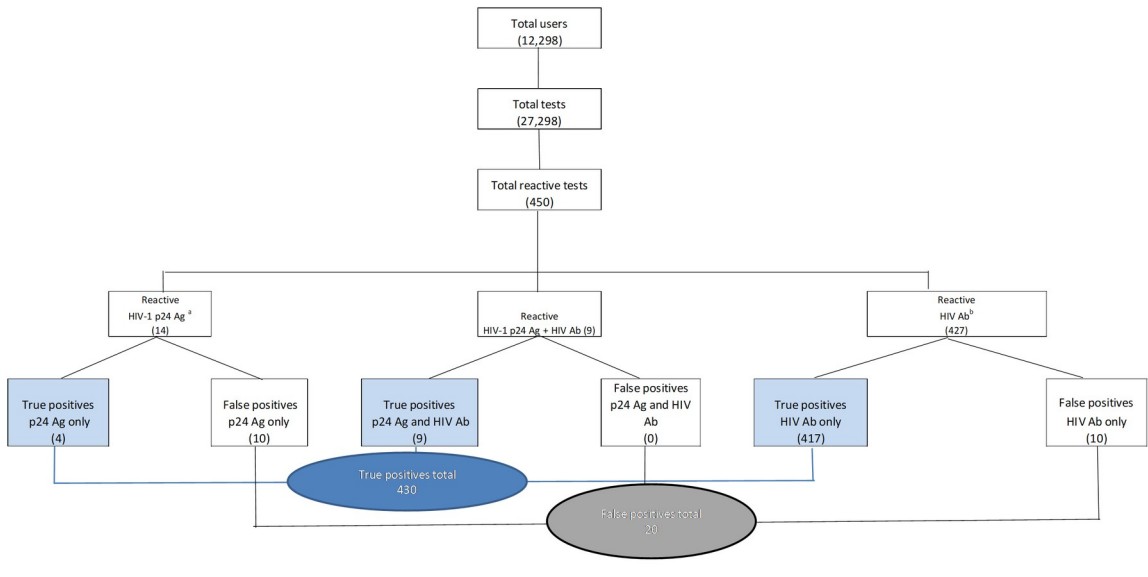

**Fig 1. Flowchart showing the results of p24 Ag and HIV Ab tests performed at BCN checkpoint.**

**Table 3. Characteristics of HIV-positive users, including antigen and/or antibody positivity, viral load, time elapsed until access to treatment; BCN checkpoint, Spain (25 March 2016 to 24 March 2019).**

| Cases | Age | p24 Ag and/or HIV Ab | Time elapsed until the first medical visit | Time elapsed until the initiation of treatment | Viral load | | CD4 | Western blot | |
|---|---|---|---|---|---|---|---|---|---|
| | | | | | Copies/ml | Log₁₀ | | Result | Band profile |
| No. 1 | 35 | p24 Ag only | 1 day | Within the same day | >10 million | >7.00 | 257 | NA | |
| No. 2 | 46 | p24 Ag only | 1 day | Within the same day | >10 million | >7.00 | 332 | Indeterminate | GP41 |
| No. 3 | 60 | p24 Ag only | 2 days | Within the same day | 4,598,724 | 6.66 | 385 | Negative | |
| No. 4 | 28 | p24 Ag only | 3 days | Within the same day | >10 million | >7.00 | 200 | Negative | |
| No. 5 | 30 | p24AG and HIV Ab | 3 days | 27 days | 186,395 | 5.27 | 475 | Positive | GP120, GP41, P24, P17 |
| No. 6 | 45 | p24AG and HIV Ab | 16 days | Within the same day | 565,702 | 5.75 | 301 | Positive | GP41, P24 |
| No. 7 | 22 | p24AG and HIV Ab | 12 days | 3 days | 270,345 | 5.43 | 144 | Positive | GP120, GP41, P31, P24 |
| No. 8 | 28 | p24AG and HIV Ab | 5 days | 2 days | 240,892 | 5.38 | 238 | Positive | GP120, GP41, P24 |
| No. 9 | 21 | p24AG and HIV Ab | 7 days | 6 days | 183,819 | 5.26 | 103 | Positive | GP120, GP41, P24 |
| No. 10 | 40 | p24AG and HIV Ab | 1 day | Within the same day | 942,000 | 5.97 | 39 | Positive | GP120, GP41, P24 |
| No. 11 | 39 | p24AG and HIV Ab | 7 days | Within the same day | 83,250 | 4.92 | 610 | Positive | GP120, GP41, P24 |
| No. 12 | 36 | p24AG and HIV Ab | 3 days | Within the same day | 300,000 | 5.48 | 559 | Positive | GP120, GP41, P24, P31, P17 |
| No. 13 | 20 | p24AG and HIV Ab | NA | NA | NA | NA | 544 | Positive | GP120, GP41, P24, P31, P17 |

(+) Positive result; (-) Negative result; NA: not available.

As for the status of the immune system at diagnosis, in the "p24 Ag only" group three out of the four cases (75.00%) were diagnosed with CD4 counts lower than 350 cells/μl, which are being classified in HIV reporting as late presenters, while being clearly acute HIV infections. Similarly, in the group of "both p24 Ag and HIV Ab" three out of nine diagnosis (33.33%) presented CD4 counts under 200 cells/μl, classified as advanced disease, and two out of nine (22.22%) ranged between 200–350 cells/μl (late presenters).

## Discussion

All participants included in this study were MSM without a previous HIV diagnosis. The study included a high number of clients and tests (27,298 tests for 12,691 clients during three years). The increase in testing frequency observed during the period, as well as the incorporation of a fourth-generation rapid test, showed a detection of confirmed HIV cases with a prevalence of 3.32%. It is also worth mentioning that the recommendation for regular testing allowed the identification of four cases of acute HIV infections ("p24 Ag only") and nine others with detectable p24 Ag and HIV Ab. All detected cases benefited from early treatment, the four with AHI within the first 72 hours after diagnosis, coinciding with the peak of HIV transmissibility, for which many new transmissions were avoided and, therefore, could be considered a high added value for prevention.

Furthermore, current widespread recommendation regarding regular HIV testing, which BCN Checkpoint first implemented in Barcelona in 2014, was welcomed and adopted by the

community. Those with a higher number of sexual partners had shortened the period to make a next appointment for HIV testing comparing to those with fewer sexual partners; to adopt regular HIV testing within the sexual health policy could reflect a better understanding by the community of the importance of preventive measures and leading to a responsible individual attitude. This is important to highlight for MSM prevention strategies as fast treatment initiation after a recent infection both improve quality of life of the individual as well as diminish HIV incidence by interrupting the transmission chain within the community [25].

On the other hand, in the "p24 Ag only" group 10 false-positive results were observed. Although investigation concluded that these cases were within the normal range, providers should take into account that such result may induce considerable anxiety for the person involved, particularly if confirmatory test result take several days to be delivered. However, in this context the PoC HIV RNA test was used which gave the final result within 90 minutes, thus minimising the impact on well-being of the person. Nevertheless, the algorithm of confirmatory testing should be reviewed as for acute HIV infection cases.

The highest HIV prevalence—close to 4%—is found in the 25 to 34 age-group. In addition, this age group accounts for half of the infections. HIV infection is commonly observed in young age-groups, as seen in previous studies [26]. A study in 2018 estimated the average age of people detected with HIV infection to be 35 years [2], while our study found the average age for receiving an HIV-positive diagnosis to be 33.22 years. This may be due to different time periods between infection and detection. Also, detection of very early infections and the identification of the at-risk groups offer us the opportunity to raise awareness about HIV infection risk and the benefits of entering a HIV Pre-Exposure Prophylaxis (PrEP) programme.

Alere™ HIV Combo (Abbott) test shortens the time to diagnosis of HIV infection and the p24 Ag detects the virus at its peak viral load and transmission capacity [27–29]. Likewise, an increase in detection of 0.94% of diagnoses has been achieved (four cases), which has great value for Public Health. Also, the viral load at the moment of detection for those with "p24 Ag only" was at least one logarithm higher than those with infections presenting with HIV Ab. Timely detection of the p24 Ag positive cases, as well as immediate treatment initiation, greatly reduces new infections, as infections at this stage are 9–10 times more transmissible than at later stages [7].

## Cost-effectiveness estimation

**Public health implications.** The reproduction rate (R0) varies throughout the course of HIV infection, but it is common knowledge that the period of peak transmission occurs after primo-infection, when viral load is at its highest. Rapid diagnosis and prompt treatment initiation of the four acute infection cases in our study contributed to a decrease in new infections. If these detections were missed at this stage, they may have been detected months later, resulting in many new transmission cases. This is a public health concern because, as seen in the United States, early detection and treatment can reduce transmission rates down to 50%, which may result in an R0 lower than a unit. This may result, ultimately, in the elimination of HIV [30].

**Economic implications.** Regarding the price of rapid testing, the average cost of an anti-HIV Ab test (third-generation test) available in the Spanish market can be estimated at 3.00 euro, and the test including p24 Ag (fourth-generation test) at average around 5.00 euro. Therefore, we can calculate from here that performing 27,298 tests including p24 Ag has resulted in an additional cost of 54,596.00 euro. In other words, every AHI case diagnosed, with at least a 1-log higher viral load, compared to other cases, has resulted in an additional surcharge of 13,649.00 euro per case.

A cost-effectiveness analysis, which was not the purpose of this study, should be performed to assess the additional surcharge of fourth-generation tests in relation to the total cost (either direct or indirect) of avoiding new infections. Nonetheless, given the simple procedure of performing the test and its contribution to the detection of AHI cases, we estimate that its inclusion for screening in key populations is an added value in the test-and-treat strategy [5].

However, some limitations in this study should be noted. First, prevalence figures refer to the population that visit BCN Checkpoint and cannot be extrapolated to other key populations or general population. Second, we were not able to identify HIV subtypes of confirmed HIV cases.

It should be noted that the study included a large number of clients (12,961 during the three-year period) and tests performed (27,298), which allowed obtainable and reliable results of the use of fourth-generation test in at-risk populations for detecting AHI.

In summary, strategies based on promoting routine testing in a community centre for men who have sex with men and transgender people contribute to increase awareness regarding HIV transmission. Detection of acute and early HIV infection through a rapid test, rapid confirmation and fast initiation of treatment allow conservation of the individual immune system and contribute to reducing HIV incidence and meet UNAIDS goals by 2030.

## Supporting information

**S1 Data.**
(XLSX)

## Acknowledgments

The authors acknowledge the membership of the BCN Checkpoint Working Group for the collaboration with the research activity, planning and execution: Albert Alonso, Jordi Martínez, Javier Fernández, Àngel Rivero, Johann Alexander Ruíz, Joan Reguant, Andreu Llorca, Daniel Michael Jacobs, Eric Muñoz, Javier Sotomayor, Francisco Áñez, Emili Aldabó, Iñaki Barquín, Sergio Cazorla, Toni Feixa, Jean Michel Fuertes, Eric Galvé, Pedro Gutiérrez, Rubén López, Óscar Martínez, Gonzalo Martínez, Manuel Mazarío, Joan Francesc Mir, Miquel Mochales, David Palma, Joel Pantaleón, Aniol Oliver, Ricard Samitier, Mario Ristovski.

## Author Contributions

**Conceptualization:** Jorge Saz, Albert Dalmau-Bueno, Michael Meulbroek, Ferran Pujol.

**Data curation:** Albert Dalmau-Bueno, Michael Meulbroek, Félix Pérez, Giovanni Marazzi.

**Formal analysis:** Jorge Saz, Albert Dalmau-Bueno, Michael Meulbroek.

**Funding acquisition:** Michael Meulbroek.

**Investigation:** Jorge Saz, Albert Dalmau-Bueno, Michael Meulbroek, Ferran Pujol, Josep Coll.

**Methodology:** Jorge Saz, Albert Dalmau-Bueno, Michael Meulbroek, Ferran Pujol, Josep Coll, Félix Pérez.

**Project administration:** Jorge Saz.

**Supervision:** Jorge Saz, Michael Meulbroek, Ferran Pujol, Josep Coll.

**Validation:** Félix Pérez.

**Writing – original draft:** Jorge Saz, Albert Dalmau-Bueno, Michael Meulbroek, Ferran Pujol, Dante R. Culqui, Joan A. Caylà.

**Writing – review & editing:** Jorge Saz, Albert Dalmau-Bueno, Michael Meulbroek, Ferran Pujol, Josep Coll, Ángel Herraiz-Tomey, Félix Pérez, Giovanni Marazzi, Héctor Taboada, Dante R. Culqui, Joan A. Caylà.

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
