## [Decision Letter · Decision Letter 0]

1 Apr 2021

PONE-D-21-05973

Use of fourth-generation rapid combined antigen and antibody diagnostic tests for the detection of acute HIV infection in a community centre for men who have sex with men, between 2016 and 2019

PLOS ONE

Dear Dr. Berges,

Thank you for submitting your manuscript to PLOS ONE. After careful consideration, we feel that it has merit but does not fully meet PLOS ONE’s publication criteria as it currently stands. Therefore, we invite you to submit a revised version of the manuscript that addresses the points raised during the review process.

We look forward to receiving your revised manuscript.

Kind regards,

Justyna Dominika Kowalska

Academic Editor

PLOS ONE

Journal Requirements:

"Abbott Laboratories has financed the editing and publishing of the results. However,

the company had no role in study design, data collection and analysis, decision to

publish, or preparation of the manuscript."

We note that you received funding from a commercial source: "Abbott Laboratories"

4. One of the noted authors is a group or consortium "BCN Checkpoint Working Group.". In addition to naming the author group, please list the individual authors and affiliations within this group in the acknowledgments section of your manuscript. Please also indicate clearly a lead author for this group along with a contact email address.

Reviewers' comments:

Reviewer's Responses to Questions

**Comments to the Author**

1. Is the manuscript technically sound, and do the data support the conclusions?

Reviewer #1: Yes

Reviewer #2: Yes

2. Has the statistical analysis been performed appropriately and rigorously? 

Reviewer #1: Yes

Reviewer #2: Yes

3. Have the authors made all data underlying the findings in their manuscript fully available?

Reviewer #1: Yes

Reviewer #2: Yes

4. Is the manuscript presented in an intelligible fashion and written in standard English?

Reviewer #1: Yes

Reviewer #2: Yes

5. Review Comments to the Author

Reviewer #1: The issue contained in the article presents the activity of the BCN Checkpoint, which is promoting sexual health among men who have sex with men (MSM) and focuses on the early diagnosis of HIV and the earliest possible initiation of cART. The fourth-generation rapid HIV tests were used for testing, allowing the simultaneous detection of p24 Ag and HIV Ab. This is important for the early detection of HIV infection. The study analyzed three possible groups of positive results and low percentage of people with false positive results. People with confirmed HIV infection were referred to the HIV unit to comence cART on the next working day. The necessity of repeating the test for those who have teted negative was emphasized and attention was drawn to the prompt initiation of cART, which contributes to the improvement of human health and reduces the HIV transmission chain.

The issues contained in the article present the current, especially in the current epidemiological situation.

Manuscript is valuable and well fitted for publication in „PLOS One”.

Reviewer #2: 1. The study that does identify an important issue which is diagnosing acute HIV infection, include numerous group of participations with data that supports the conclusions.

2.The statistical analysis has been performed appropriately.

3. The authors made all data underlying the findings in their manuscript fully available without restriction.

4. The paper is generally well written and structured. Overall structure of the article is well organized and well balanced.

Comments:

Line 53

Add „combined” to anitiretroviral treatment so that it matches the abbreviation - cART

Line 76

Use more formal language

Line 115

Full name should be ‘infectious mononucleosis’. Change common name ‘flu’ to infuenza.

Line 324

There no necessarily to extract gay men from others MSM.

6. PLOS authors have the option to publish the peer review history of their article (what does this mean?). If published, this will include your full peer review and any attached files.

Reviewer #1: No

Reviewer #2: No

---

## [Author Response · Author response to Decision Letter 0]

29 May 2021

Thank you for the review and please find below the responses to all issues that were raised:

1. Ensure that your manuscript meets PLOS ONE's style requirements.

Response: Manuscript has been adapted to style requirements.

2. Review your reference list.

Response: Reference list has been checked.

3. Provide an amended Competing Interests Statement.

Response: The updated Competing Interests Statement has been included in the cover letter.

4. List the individual authors and affiliations within this BCN Checkpoint Working Group.

Response: Individual authors and affiliations have been updated.

5. Reviewers' comments:

1. Is the manuscript technically sound, and do the data support the conclusions?

Reviewer #1: Yes

Reviewer #2: Yes

2. Has the statistical analysis been performed appropriately and rigorously? 

Reviewer #1: Yes

Reviewer #2: Yes

3. Have the authors made all data underlying the findings in their manuscript fully available?

Reviewer #1: Yes

Reviewer #2: Yes

4. Is the manuscript presented in an intelligible fashion and written in standard English?

Reviewer #1: Yes

Reviewer #2: Yes

5. Review Comments to the Author

Reviewer #1: The issue contained in the article presents the activity of the BCN Checkpoint, which is promoting sexual health among men who have sex with men (MSM) and focuses on the early diagnosis of HIV and the earliest possible initiation of cART. The fourth-generation rapid HIV tests were used for testing, allowing the simultaneous detection of p24 Ag and HIV Ab. This is important for the early detection of HIV infection. The study analyzed three possible groups of positive results and low percentage of people with false positive results. People with confirmed HIV infection were referred to the HIV unit to comence cART on the next working day. The necessity of repeating the test for those who have teted negative was emphasized and attention was drawn to the prompt initiation of cART, which contributes to the improvement of human health and reduces the HIV transmission chain.

The issues contained in the article present the current, especially in the current epidemiological situation.

Manuscript is valuable and well fitted for publication in „PLOS One”.

Response: Thank you.

Reviewer #2: 1. The study that does identify an important issue which is diagnosing acute HIV infection, include numerous group of participations with data that supports the conclusions.

2.The statistical analysis has been performed appropriately.

3. The authors made all data underlying the findings in their manuscript fully available without restriction.

4. The paper is generally well written and structured. Overall structure of the article is well organized and well balanced.

Comments:

Line 53

Add „combined” to anitiretroviral treatment so that it matches the abbreviation – cART

Response: The word “combined” has been added.

Line 76

Use more formal language

Response: The phrase has been rewritten as requested.

Line 115

Full name should be ‘infectious mononucleosis’. Change common name ‘flu’ to infuenza.

Response: The terms have been changed.

Line 324

There no necessarily to extract gay men from others MSM.

Response: The phrase has been rewritten as requested.

Kind regards,

Jorge Saz Berges

---

## [Decision Letter · Decision Letter 1]

9 Jul 2021

Use of fourth-generation rapid combined antigen and antibody diagnostic tests for the detection of acute HIV infection in a community centre for men who have sex with men, between 2016 and 2019

PONE-D-21-05973R1

Dear Dr. Berges,

We’re pleased to inform you that your manuscript has been judged scientifically suitable for publication and will be formally accepted for publication once it meets all outstanding technical requirements.

Kind regards,

Justyna Dominika Kowalska

Academic Editor

PLOS ONE

Additional Editor Comments (optional):

Reviewers' comments:

Reviewer's Responses to Questions

**Comments to the Author**

1. If the authors have adequately addressed your comments raised in a previous round of review and you feel that this manuscript is now acceptable for publication, you may indicate that here to bypass the “Comments to the Author” section, enter your conflict of interest statement in the “Confidential to Editor” section, and submit your "Accept" recommendation.

Reviewer #2: All comments have been addressed

2. Is the manuscript technically sound, and do the data support the conclusions?

Reviewer #2: Yes

3. Has the statistical analysis been performed appropriately and rigorously? 

Reviewer #2: Yes

4. Have the authors made all data underlying the findings in their manuscript fully available?

Reviewer #2: Yes

5. Is the manuscript presented in an intelligible fashion and written in standard English?

Reviewer #2: Yes

6. Review Comments to the Author

Reviewer #2: (No Response)

7. PLOS authors have the option to publish the peer review history of their article (what does this mean?). If published, this will include your full peer review and any attached files.

Reviewer #2: No

---

## [Editor Report · Acceptance letter]

16 Jul 2021

PONE-D-21-05973R1 

Use of fourth-generation rapid combined antigen and antibody diagnostic tests for the detection of acute HIV infection in a community centre for men who have sex with men, between 2016 and 2019 

Dear Dr. Saz:

I'm pleased to inform you that your manuscript has been deemed suitable for publication in PLOS ONE. Congratulations! Your manuscript is now with our production department. 

Kind regards, 

on behalf of

Dr. Justyna Dominika Kowalska 

Academic Editor

PLOS ONE